# Characterization of Antimicrobial Compounds from *Trichoderma flavipes* Isolated from Freshwater Environments

**DOI:** 10.3390/jof11120857

**Published:** 2025-12-02

**Authors:** Jeong Tae Kim, Won Su Cheon, Sanghee Lee, Jaeduk Goh, Chang Soo Lee, Hye Yeon Mun

**Affiliations:** Biological Resources Research Department, Nakdonggang National Institute of Biological Resources, Sangju 37242, Republic of Korea

**Keywords:** antimicrobial, freshwater fungi, secondary metabolites, *Trichoderma flavipes*

## Abstract

Traditional antibiotic agents are commonly employed in the control of pathogenic microorganisms. However, there is a growing need for novel alternative antimicrobial agents owing to the increasing prevalence of resistance to these treatments. Freshwater fungi, recognized for producing diverse secondary metabolites with biological activities, offer promising sources for drug development. However, studies on *Trichoderma flavipes* remain limited. Therefore, this study was conducted to investigate the antimicrobial properties of bioactive compounds derived from *Trichoderma flavipes* FBCC-F1632, a fungal species isolated from freshwater environments in Korea. The fungal strain FBCC-F1632 was isolated from stream soil obtained from Mungyeon-si, Republic of Korea, and identified through DNA extraction and phylogenetic analysis. Antimicrobial activity against *Staphylococcus aureus* and *Bacillus cereus* was assessed, revealing significant inhibitory rates. Potential bioactive compounds were extracted, purified, and structurally characterized using chromatographic and spectroscopic techniques, including nuclear magnetic resonance and mass spectrometry. Five bioactive compounds were identified: F1632-1 (cordyol C), F1632-2 (diorcinol), F1632-3 (violaceol I), F1632-4 (tryptophol), and F1632-6 (violaceol II). These compounds exhibited notable antimicrobial activities, particularly against *Staphylococcus aureus* and *Bacillus cereus*, underscoring their potential as candidates for the development of novel antimicrobial therapeutics.

## 1. Introduction

Freshwater fungi are a group of microorganisms that play a crucial role in the ecological dynamics of freshwater ecosystems such as rivers, streams, and lakes. They function as key decomposers of phytoplankton and other organic matter in these environments. To date, approximately 3870 freshwater fungal species have been identified [1]. These fungi have developed unique metabolic pathways to adapt to various environmental stresses, resulting in the production of diverse secondary metabolites [2]. These metabolic products are not only essential for their survival and growth but also exhibit various biological activities [2]. Research has shown that freshwater fungi can produce compounds with physiological activities, such as antibacterial, antifungal, antiviral, and anticancer activities, rendering them potential candidates for drug development. Although fungi are widely acknowledged as a prolific source of bioactive secondary metabolites, freshwater fungi have been relatively underexplored in this regard [2,3]. Research on the metabolic products of freshwater fungi remains in its early stages and is significantly less advanced compared to studies on marine and terrestrial fungi. However, interest in the metabolites produced by freshwater fungi has grown in recent years, and their potential medicinal applications are being actively investigated [3,4].

The emergence and spread of microorganisms have become a significant public health concern in contemporary society. Pathogenic microorganisms, in particular, pose a significant threat to human health by causing infectious diseases [5]. Consequently, the effective suppression and management of these microorganisms have become crucial areas of research and development. Although traditional antibiotics and antifungal agents are widely used, the growing prevalence of resistance to these treatments highlights the urgent need for novel alternative compounds [6].

*Trichoderma flavipes* is a species within the phylum Ascomycota, class Sordariomycetes, order Hypocreales, and genus *Trichoderma*. The genus *Trichoderma* is recognized for its notable biological and ecological significance [7,8]. Species of this genus produce antibiotics that inhibit the growth of other microorganisms, including fungi, bacteria, and viruses, with a particular efficacy for suppressing plant pathogens [9]. Additionally, *Trichoderma* species are known to secrete plant growth hormones, such as auxins and cytokinins, and produce enzymes, such as cellulase, chitinase, and protease [9,10,11]. Owing to these beneficial properties, the genus *Trichoderma* has become an important focus of ongoing research in life sciences and industrial applications. However, *T. flavipes* has not been extensively studied. To date, cordyol C, diorcinol, violaceol I/II, and tryptophol have been isolated from various fungal taxa such as *Aspergillus* and *Emericella* spp., but their occurrence in *Trichoderma flavipes* has not been previously reported. Therefore, this study represents the first identification of these metabolites from *T. flavipes*, highlighting the species as a novel source of antimicrobial compounds [12,13,14].

This study aimed to explore and analyze the chemical properties of metabolites produced by the freshwater fungus *Trichoderma flavipes* FBCC-F1632 and assess its antibacterial activities.

## 2. Materials and Methods

### 2.1. Isolation and Identification of FBCC-F1632

The fungal strain FBCC-F1632 was isolated from a freshwater sample collected from Kimyong-ri (36°44′26.1″ N, 128°13′19″ E), Sanbuk-myeon, Mungyeong-si, Republic of Korea, during the summer of July 2021. At the sampling site, a 50 mL sample of freshwater was filtered directly using a hand pump and an MCE membrane filter (HAWP04700, MF-Millipore™, Tullagreen, Ireland). Subsequently, the filter paper was placed on water agar plates containing 20 g/L agar (Difco; BD, Franklin Lakes, NJ, USA), 1 L distilled water, and 100 ppm streptomycin (Sigma, Darmstadt, Germany) to inhibit bacterial growth. The samples were transported to the laboratory and incubated at 15 °C to promote spore isolation. Single spores were isolated, and purified fungal cultures were transferred to potato dextrose agar (PDA; Difco) and incubated at 20 °C to establish pure cultures. All fungal strains used in this study were maintained on PDA at 25 °C. Morphological characteristics were observed using an Eclipse Ni-U microscope (Nikon, Tokyo, Japan).

### 2.2. DNA Extraction and Phylogenetic Analysis

Fungal genomic DNA was extracted using the NucleoSpin Plant II DNA extraction kit (Macherey-Nagel, Duren, Germany) according to the manufacturer’s instructions. To identify the fungal species, the RNA polymerase II (*RPB2*) gene was targeted and amplified using specific primers 5f (5′-GAYGAYMGWGATCAYTTYGG-3′) and 7cr (5′-CCCATRGCTTGYTTRCCCAT-3′) [15]. DNA sequence homology was assessed by performing BLAST searches on the National Center for Biotechnology Information (NCBI) database to compare the sequences with existing records. For phylogenetic analysis, MEGA11 software [16] was used to construct and analyze the evolutionary relationships among the fungal strains. A phylogenetic tree was generated using the neighbor-joining method with 1000 bootstrap replications. This analysis included 36 nucleotide sequences. All positions with less than 95% site coverage were eliminated, indicating that fewer than 5% alignment gaps, missing data, or ambiguous bases were allowed at any position (partial deletion option). The final dataset comprised 889 positions. Reference sequences for other fungi were obtained from GenBank at NCBI.

### 2.3. Antimicrobial Activity

The bacterial strains used in this study, *Staphylococcus aureus* CCARM3089 (methicillin-resistant strain) and *Bacillus cereus* CCARM0120, were obtained from the Culture Collection of Antimicrobial Resistant Microbes (CCARM, Seoul, Republic of Korea). These strains are known for their resistance to several antibiotics, making them suitable models for evaluating new antimicrobial agents. A disk diffusion assay was conducted to evaluate the antibacterial activity against *S. aureus* CCARM3089 and *B. cereus* CCARM0120. The overlay medium was prepared by pouring 20 mL of R2A agar (Difco) containing a 5% suspension of *S. aureus* and *B. cereus* (adjusted to OD_600_ = 0.5) into 9 cm Petri dishes and allowing it to solidify. Next, 50 µL of the extract from *T. flavipes* FBCC-F1632 was carefully applied to an 8 mm sterile paper disc, which was subsequently placed at the center of the solidified overlay medium. The plates were incubated at 30 °C for 1–2 days to allow the extract to diffuse into the medium and interact with the pathogens. Minimum inhibitory concentration (MIC) and minimum bactericidal concentration (MBC) values were determined according to the Clinical and Laboratory Standards Institute (CLSI, 2021) guidelines using the broth microdilution method [17]. Streptomycin (15 µg/disc, 150 µg/disc) were used as positive controls to validate assay performance.

### 2.4. Material Separation and Purification

FBCC-F1632 culture broth (10 L) was centrifuged at 3000 rpm for 30 min at 4 °C, separating the supernatant and pellet. Next, 1 L Diaion HP-20 resin (Supelco, Darmstadt, Germany) was added to the supernatant to adsorb the active compounds. The adsorbed resin was filled into a column and eluted using 30%, 70%, and 100% methanol aqueous solutions (Thermo Fisher Scientific, Waltham, MA, USA). The fraction eluted with the 70% methanol aqueous solution, which exhibited activity, was concentrated under reduced pressure to remove the solvent. Ethyl acetate (EA, Thermo Fisher Scientific) was subsequently added for distribution extraction. The dispensed EA layer was concentrated under reduced pressure and fractionated using octadecylsilane (ODS; Sigma, Darmstadt, Germany) medium-pressure liquid chromatography (MPLC). ODS MPLC was conducted using Combiflash RF+ (Teledyne ISCO, Lincoln, NE, USA) equipment, employing deionized water-methanol as a mobile phase and a C18 column (86 g, Teledyne ISCO, Lincoln, NE, USA). Based on the ODS MPLC analysis, the compound was separated and purified by dividing it into four groups (fraction [Fr.] 1, Fr. 2, Fr. 3, and Fr. 4), followed by Sephadex LH-20 column chromatography (Supelco, Darmstadt, Germany). The entire separation process is schematically shown in Figure 1.

### 2.5. Structure Analysis

Electrospray ionization (ESI) mass spectra were obtained using an ESI-QTRAP 3200 mass spectrometer (Applied Biosystems, Thermo Fisher Scientific), whereas NMR spectra were recorded on a JEOL JNM-ECZ500R 500 MHz FT-NMR spectrometer (JEOL Ltd., Tokyo, Japan) operating at 500 MHz for ^1^H NMR and 125 MHz for ^13^C NMR, using CD_3_OD (deuterated methanol, Thermo Fisher Scientific) as the solvent. Chemical shifts are reported in parts per million (ppm, δ) relative to tetramethylsilane (TMS) as the internal standard. For NMR analysis, both two-dimensional (^1^H–^1^H COSY, heteronuclear single quantum coherence (HMQC), and heteronuclear multiple bond correlation (HMBC)) and one-dimensional (^1^H and ^13^C) spectra were acquired. NMR data were interpreted according to standard spectral analysis protocols.

## 3. Results

### 3.1. Isolation and Identification of FBCC-F1632

The unique morphological and genetic characteristics of FBCC-F1632 provided valuable insights into its classification and ecological adaptability. The dual-colony morphology observed in FBCC-F1632, with growth patterns resembling those of both *Trichoderma* and *Stilbella* species, suggests a possible adaptive mechanism that enables this isolate to thrive under varying environmental conditions. This adaptability is particularly noteworthy, given that *Trichoderma* species are generally known for their rapid growth, whereas *Stilbella* species exhibit slower, more conservative growth patterns [18]. Such morphological variability could be a response to differing nutrient availability or environmental pressures in freshwater ecosystems (Figure 2a,b).

The conidial morphology of FBCC-F1632, characterized by globose conidia and penicillate, slender phialides, further supports its close relationship with *T. flavipes* (Figure 2c). These morphological features, coupled with the high sequence similarity (99.10%) to *T. flavipes* based on the *RPB2* gene, firmly place FBCC-F1632 within the *Trichoderma* clade (Figure 3). The genetic similarity also suggests that FBCC-F1632 may share biological or ecological functions with *T. flavipes*, such as antifungal or antibacterial activities, which are commonly observed in many *Trichoderma* species known for their biocontrol properties [19].

Phylogenetic note: phylogenetic analysis revealed a consistency index of 0.663194 (0.630476), a retention index of 0.819535 (0.819535), and a composite index of 0.543511 (0.516697) for all sites and parsimony-informative sites (in parentheses). The percentage of replicate trees in which the associated taxa clustered together in the bootstrap test (1000 replicates) is indicated next to the branches [20]. The maximum parsimony tree was constructed using the tree-bisection-regrafting algorithm [21] with a search level of 2, where initial trees were generated by the random addition of sequences (10 replicates). This analysis involved 20 nucleotide sequences, with a total of 858 positions in the final dataset. Evolutionary analyses were conducted using MEGA11 [16].

### 3.2. Antimicrobial Activity of FBCC-F1632

This study investigates the antimicrobial efficacy of *T. flavipes* FBCC-F1632 against two clinically relevant bacterial strains, *Staphylococcus aureus* CCARM3089 and *Bacillus cereus* CCARM0120. As shown in Figure 4, clear inhibition zones were observed around FBCC-F1632 discs (22.33 ± 0.31 mm for *S. aureus* and 12.67 ± 0.58 mm for *B. cereus*), comparable to those produced by streptomycin (15 µg/disc and 150 µg/disc). These findings indicate that *T. flavipes* produces metabolites capable of suppressing antibiotic-resistant strains.

The ability of *T. flavipes* to inhibit antibiotic-resistant *S. aureus* expands its potential applications, extending beyond agriculture to areas such as food safety and potentially clinical settings, where resistant infections pose substantial health risks. Identifying and characterizing the specific compounds responsible for this activity could pave the way for developing alternative antimicrobial agents or supplements to existing antibiotics. Additionally, investigating whether these compounds are effective against other resistant strains could provide a broader understanding of the antibacterial spectrum of *T*. *flavipes*.

### 3.3. Compound Extraction from FBCC-F1632

Bioactive compounds were successfully extracted from FBCC-F1632 and fractionated, highlighting the potential of this isolate as a source of novel metabolites. A combination of EA extraction, ODS MPLC, and Sephadex LH-20 column chromatography facilitated the effective separation and isolation of distinct compounds, representing essential steps in identifying and characterizing bioactive molecules. Specific compounds were isolated in varying quantities, such as F1632-2 (58.0 mg), the most abundant compound, and F1632-6 (3.2 mg), which was further purified using preparative high-performance liquid chromatography (HPLC). These results suggest a wide range of metabolite diversity within FBCC-F1632. The successful isolation of F1632-6 in pure form is particularly significant, as it can facilitate its subjection to further structural and bioactivity studies to assess its potential applications. These methodologies, including fractionation by ODS MPLC and final purification through HPLC, highlight the importance of multistep chromatographic processes for accessing pure forms of bioactive metabolites from complex fungal extracts.

### 3.4. Structure Analysis

#### 3.4.1. Structure of F1632-1

The comprehensive spectral analysis of F1632-1 provided detailed information on its molecular structure and potential biological activities. ESI-MS analysis determined a molecular weight of 246, consistent with that of known metabolites in the cordycepin family. This molecular characterization is crucial for understanding the properties of the compound and its possible applications in pharmaceuticals and natural product development. The ^1^H NMR spectrum revealed distinctive peaks corresponding to aromatic methine protons and methyl groups, indicating a complex structure typical of phenolic compounds. Five aromatic methine protons were identified, suggesting a highly substituted aromatic ring, which is characteristic of compounds with potential antioxidant or antimicrobial properties. Chemical shifts and splitting patterns were observed, providing a strong basis for deducing the connectivity of the protons and the overall arrangement of the molecule. The ^13^C NMR spectrum revealed multiple oxygenated sp^2^ carbons, indicating the presence of functional groups known for their biological activities, such as hydroxyl or carbonyl groups, which may contribute to the reactivity of the compound and its interactions with biological targets (Table 1). Long-range correlations from HMQC and HMBC spectra further elucidated the structural framework, indicating a well-defined spatial arrangement, which is essential for understanding the mechanism of action of the compound (Figure 5).

#### 3.4.2. Structure of F1632-2

The spectral analysis of F1632-2 provided crucial information on its molecular structure and potential biological properties. ESI-mass analysis determined a molecular weight of 230, consistent with that of known polyphenolic metabolites, which are often associated with significant bioactivity. The ^1^H NMR spectrum revealed multiple aromatic methine protons, suggesting a symmetrical aromatic system typical of polyphenols. Splitting patterns and chemical shifts were observed, indicating the electronic environment of the protons and offering insights into the connectivity within the molecule. The methyl proton peak at 2.19 ppm, with a relative intensity of 6H, indicates the presence of two equivalent methyl groups, which is a hallmark of compounds such as diorcinol. In the ^13^C NMR spectrum, oxygenated sp^2^ carbon peaks at 159.5 and 159.3 ppm were identified, indicating the presence of functional groups often implicated in antioxidant and antimicrobial activities. The distribution of sp^2^ methine and methyl carbon peaks further supports the notion of a complex molecular framework, which is critical for understanding the interactions between the compound and biological systems (Table 1). Long-range correlations were observed from the HMQC and HMBC spectra, confirming the structural integrity of F1632-2. The correlations observed between the methyl protons and various carbons within the aromatic system highlight the importance of these groups in stabilizing the overall structure through intramolecular interactions (Figure 5). This connectivity is vital for elucidating the potential mechanisms by which diorcinol exerts its biological effects.

#### 3.4.3. Structure of F1632-3

The characterization of F1632-3 as violaceol I involved a comprehensive analytical approach, highlighting the importance of employing a multi-faceted analytical approach to identify and understand natural products. ESI-mass analysis determined the molecular weight of the compound, providing a solid foundation for subsequent structural elucidation and confirming the identity of the compound [12]. The ^1^H NMR spectrum exhibited distinct patterns, indicating a well-defined structure with symmetrical features, likely contributing to the stability and bioactivity of the compound. The methyl proton peak revealed multiple methyl groups, which may influence the solubility and interactions of the compound with biological targets. In the ^13^C NMR spectrum, oxygenated sp^2^ carbons were identified, which are often associated with enhanced biological activities, including antioxidant and antimicrobial effects (Table 1) [13]. Long-range correlations from the HMQC and HMBC spectra further validated the structural assignments, providing further insights into the connectivity of the molecule and elucidating the relationships between various functional groups (Figure 5).

#### 3.4.4. Structure of F1632-4

The characterization of F1632-4 as tryptophol involved comprehensive spectroscopic analysis, highlighting the importance of employing detailed spectroscopic analysis to elucidate the structures of natural products. ESI-mass analysis determined the molecular weight of the compound, providing a solid foundation for subsequent structural elucidation and confirming the identity of the compound. The ^1^H NMR spectrum revealed multiple aromatic methine protons, suggesting a well-defined aromatic system crucial for the biological activity typical of tryptophols. Moreover, the presence of methylene protons indicated a complex molecular framework that may facilitate interactions with biological targets. In the ^13^C NMR spectrum, sp^2^ carbon peaks were identified, indicating functional groups that may contribute to the potential bioactivity of the compound (Table 1). Long-range correlations were observed in the HMQC and HMBC spectra, providing further insights into the connectivity of the compound and supporting the structural assignments made based on the proton and carbon chemical shifts (Figure 5).

#### 3.4.5. Structure of F1632-6

The characterization of F1632-6 as violaceol II involved detailed spectroscopic analysis, highlighting the importance of comprehensive spectroscopic analysis for the structural elucidation of natural products. ESI-mass analysis determined the molecular weight of the compound, providing a strong basis for subsequent structural interpretations. The ^1^H NMR spectrum revealed several aromatic methine protons, indicating a robust aromatic system critical for biological activity typical of compounds such as violaceol II [18]. The presence of multiple methyl groups suggests that these structural elements may enhance the lipophilicity of the compound, potentially influencing its bioavailability and interaction with biological targets. In the ^13^C NMR spectrum, sp^2^ carbon peaks were detected, confirming the presence of functional groups relevant to the known properties of violaceol II (Table 1) [13]. Long-range correlations were observed in the HMQC and HMBC spectra, providing further insights into the connectivity of the compound and corroborating the assignments made based on the chemical shifts (Figure 5).

### 3.5. Antimicrobial Activity of Compounds Isolated from FBCC-F1632

The antimicrobial activities of five isolated compounds (F1632-1 to F1632-5) were evaluated against Staphylococcus aureus and Bacillus cereus using minimum inhibitory concentration (MIC) and minimum bactericidal concentration (MBC) assays (Table 2). The selection of these bacterial strains is particularly relevant, as both strains are known to cause serious infections in humans and are associated with various foodborne illnesses [22,23].

Among the tested compounds, Violaceol I (F1632-3) exhibited the lowest MIC value against *S. aureus* (12.5 µg/mL), indicating the highest inhibitory potency toward this strain. For *B. cereus*, Cordyol C (F1632-1) demonstrated the lowest MIC (12.5 µg/mL), followed by Violaceol II (F1632-5) and Violaceol I (F1632-3) with MIC values of 25 µg/mL.

MBC values were generally higher or equal to the MICs, with most compounds showing an MBC of 50 µg/mL against both bacterial species. Notably, Cordyol C displayed a fourfold difference between MIC and MBC against *B. cereus* (12.5 µg/mL vs. 50 µg/mL), suggesting a primarily bacteriostatic effect at lower concentrations.

## 4. Discussion

This study investigated the antimicrobial potential of compounds isolated from *T. flavipes* FBCC-F1632, including cordyol C, diorcinol, violaceol I, tryptophol, and violaceol II. Although this strain demonstrated effectiveness against fungal pathogens under co-culture conditions, the extracted compounds did not exhibit antifungal activity in isolation. This discrepancy suggests that certain synergistic effects or environmental conditions present during co-culture may be necessary to activate or enhance the antifungal properties of these compounds. When compared with standard antibiotics, Violaceol I exhibited moderate activity against *S. aureus* (MIC 12.5 µg/mL) relative to vancomycin (MIC 1 µg/mL). Although the potency is lower than clinical antibiotics, the compound’s structural simplicity and natural origin make it a promising scaffold for chemical modification.

The identification of F1632-1 as cordyol C through database searches underscores its importance in natural product chemistry. Cordyol C, initially isolated from *Cordyceps* sp. BCC1861, exhibits diverse bioactivities, including antimicrobial, antimalarial, and cytotoxic effects, indicating its potential as a multifunctional therapeutic agent. Similarly, *Aspergillus sydowii* ZSDS1-F6, associated with marine sponges, produces various antimicrobial and antiviral sesquiterpenoids [14]. Among related compounds, cordyol E from *Aspergillus* sp. XS-20090066 has demonstrated significant antibacterial effects against pathogens, such as *Staphylococcus epidermidis*, *S. aureus*, *Vibrio anguillarum*, *Vibrio parahemolyticus*, *and Pseudomonas putida* [14,23,24]. F1632-2 was identified as diorcinol, a compound initially derived from *Aspergillus tabacinus* and *Emericella falconensis*, underscoring its relevance in natural product research [25,26,27]. Diorcinol has been demonstrated to exhibit antimicrobial properties against plant pathogenic fungi and bacteria, further supporting its potential in therapeutic applications [25,26]. Furthermore, F1632-3 and F1632-6 were identified as violaceol I and violaceol II, respectively, both of which were originally isolated from *A. tabacinus*, *Trichoderma polyalthiae*, and *Emericella violacea* [18,19,27]. These compounds exhibit a broad range of bioactivities, including antibacterial, antimalarial, anti-tuberculous, anti-herpes simplex virus type I, and cytotoxic effects [18,19,27]. The extensive bioactivities associated with these compounds highlight their potential for various medical applications, although further studies are required to elucidate their specific mechanisms of action.

Violaceol I demonstrated the strongest inhibitory effect against *S. aureus*, while Cordyol C was most effective against *B. cereus*. The relatively small gap between MIC and MBC values for most compounds suggests that bactericidal activity is achieved at concentrations close to the inhibitory threshold. Interestingly, Diorcinol (F1632-2) and Tryprophol (F1632-4) exhibited identical MIC and MBC values (50 µg/mL) for both bacterial strains, indicating moderate and non-selective antibacterial activity. In contrast, the violaceol derivatives (F1632-3 and F1632-5) showed more variable MICs, implying that subtle structural modifications may enhance or reduce potency against specific Gram-positive bacteria.

In conclusion, freshwater fungi represent a promising source of antimicrobial metabolites. Additionally, synergistic or time–kill studies could further elucidate the antimicrobial potential of these metabolites. Future research should focus on the bioprospecting of these organisms, focusing on isolating and characterizing novel compounds that can address current challenges in health and agriculture. By doing so, this study not only provides a better understanding of the complexities of microbial interactions within freshwater ecosystems but also paves the way for practical applications across various fields.

## Figures and Tables

**Figure 1 jof-11-00857-f001:**
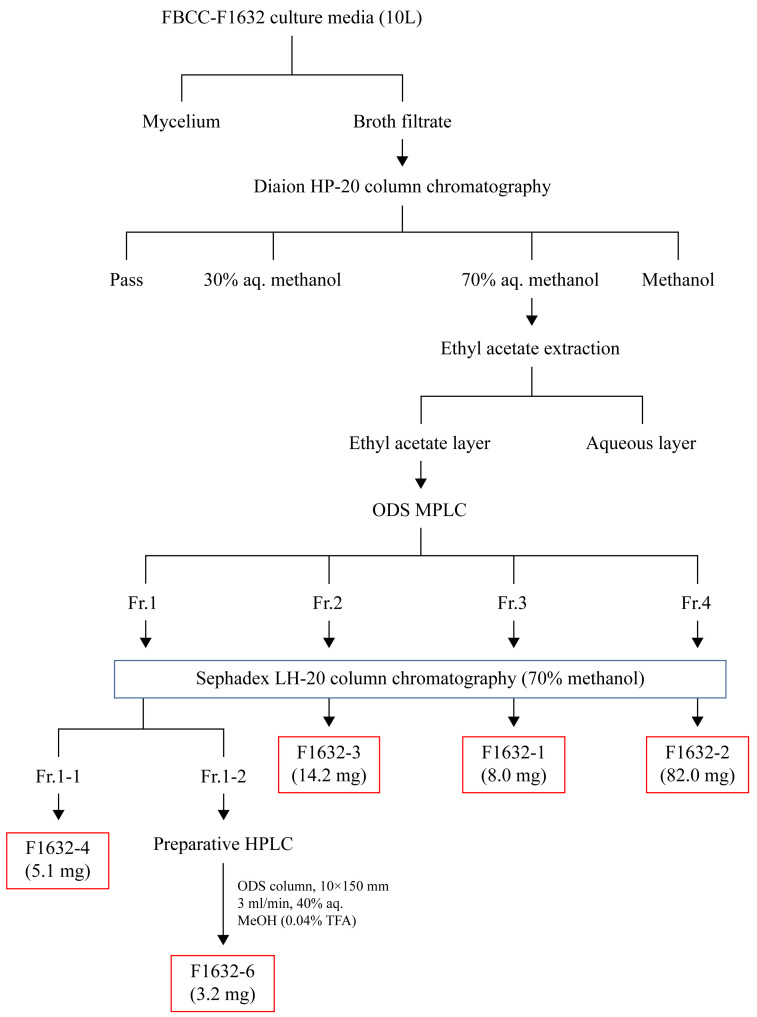
Solvent extraction of FBCC-F1632 culture media. Schematic representation of the culture media separation and purification process for F1632 (Aq: aqueous solution, Fr: fraction).

**Figure 2 jof-11-00857-f002:**
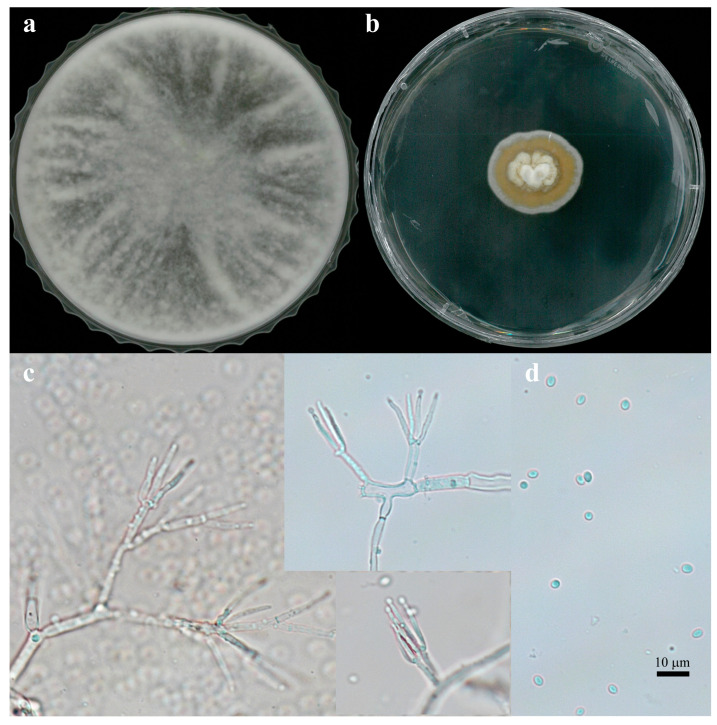
Morphological characteristics of FBCC-F1632 in culture and microscopic structures. (**a**) Colony shape resembling *Trichoderma*-like growth. (**b**) Colony shape resembling *Stilbella*-like growth. (**c**) Conidiophore and (**d**) conidia of FBCC-F1632. Scale bar = 10 µm, magnification ×400.

**Figure 3 jof-11-00857-f003:**
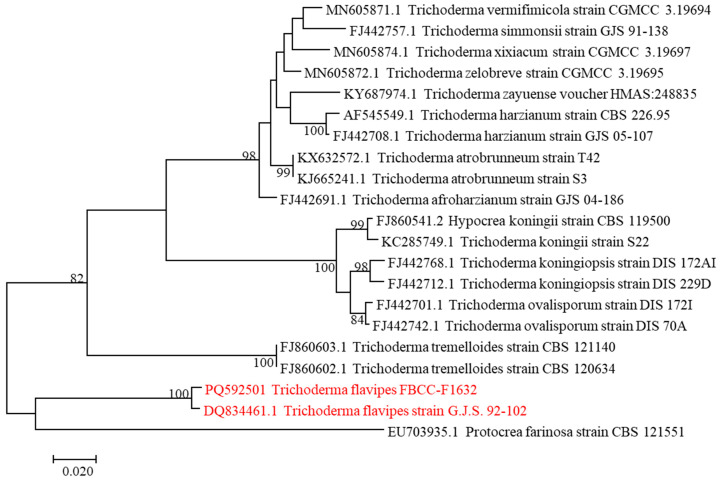
Phylogenetic tree of *Trichoderma flavipes* FBCC-F1632 (red color letter) constructed using the *RPB2* gene. The evolutionary history was inferred using the maximum parsimony method. *Protocrea farinose* served as the outgroup.

**Figure 4 jof-11-00857-f004:**
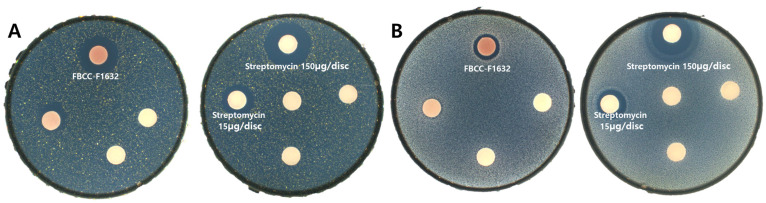
Antimicrobial activity of *T. flavipes* FBCC-F1632 against pathogenic microorganisms. (**A**) *Staphylococcus aureus* CCARM3089; (**B**) *Bacillus cereus* CCARM0120. Paper discs were loaded with FBCC-F1632 extract or streptomycin (15 µg/disc and 150 µg/disc). Clear inhibition zones indicate antibacterial activity.

**Figure 5 jof-11-00857-f005:**
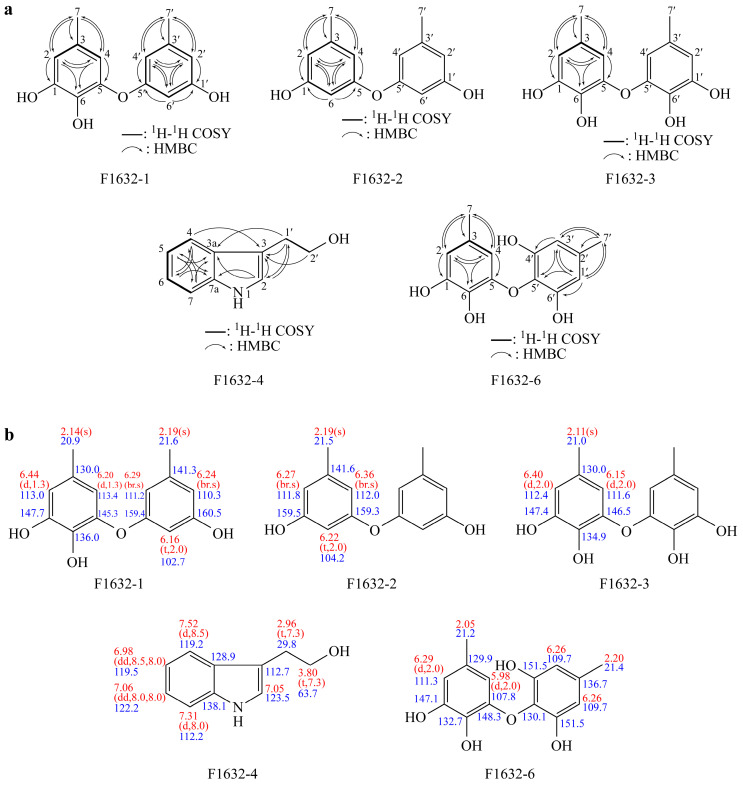
Chemical structure of isolated compounds. (**a**) Chemical structure of compounds elucidated through 2D nuclear magnetic resonance (NMR) correlation. Bold lines represent ^1^H-^1^H correlation spectroscopy analysis results, and arrows indicate heteronuclear multiple-bond correlation analysis results. (**b**) Assignments of ^1^H (red) and ^13^C (blue) NMR peaks for the elucidated compounds.

**Table 1 jof-11-00857-t001:** 1D NMR data of compounds.

Position	F1632-1		F1632-2		F1632-3		F1632-4		F1632-6	
	δ^13^C (mult.)	δ^1^H (mult.)	δ^13^C (mult.)	δ^1^H (mult.)	δ^13^C (mult.)	δ^1^H (mult.)	δ^13^C (mult.)	δ^1^H (mult.)	δ^13^C (mult.)	δ^1^H (mult.)
1	147.7		159.5		147.4				147.1	
2	113.0	6.44 (1H, d, J = 1.3 Hz)	111.8	6.27 (2H, br. s)	112.4	6.40 (2H, d, J = 2.0 Hz)	123.5	7.05 (1H, s)	111.3	6.29 (1H, d, J = 2.0 Hz)
3	130.0		141.6		130.0		112.7		129.9	
3a							128.9			
4	113.4	6.20 (1H, d, J = 1.3 Hz)	112	6.36 (2H, br. s)	111.6	6.15 (2H, d, J = 2.0 Hz)	119.2	7.52 (1H, d, J = 8.5 Hz)	107.8	5.98 (1H, d, J = 2.0 Hz)
5	145.3		159.3		146.5		119.5	6.98 (1H, dd, J = 8.5, 8.0 Hz)	148.3	
6	136.0		104.2	6.22 (2H, t, J = 2.0 Hz)	134.9		122.2	7.06 (1H, dd, J = 8.0, 8.0 Hz)	132.7	
7	20.9	2.14 (3H, s)	21.5	2.19 (6H, s)	21.0	2.11 (6H, s)	112.2	7.31 (1H, d, J = 8.0 Hz)	21.2	2.05 (3H, s)
7a							138.1			
1′	160.5						29.8	2.96 (2H, t, J = 7.3 Hz)	109.7	
2′	110.3	6.24 (1H, br. s)					63.7	3.80 (2H, t, J = 7.3 Hz)	136.7	
3′	141.3								109.7	6.26 (2H, s)
4′	111.2	6.29 (1H, br. s)							151.5	
5′	159.4								130.1	
6′	102.7	6.16 (1H, t, J = 2.0 Hz)							151.5	
7′	21.6	2.19 (3H, s)							21.4	2.20 (3H, s)

**Table 2 jof-11-00857-t002:** MIC and MBC test results of the compounds against pathogenic bacteria.

No.	Compound	Name	MIC		MBC	
			*S. aureus*	*B. cereus*	*S. aureus*	*B. cereus*
1	F1632-1	Cordyol C	50	12.5	50	25
2	F1632-2	Diorcinol	50	25	-	50
3	F1632-3	Violaceol I	12.5	25	25	50
4	F1632-4	Tryptophol	-	-	-	-
5	F1632-6	Violaceol II	25	25	50	50

## Data Availability

The original contributions presented in this study are included in the article. Further inquiries can be directed to the corresponding author.

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
