# Peer review of "Characterization of Antimicrobial Compounds from Trichoderma flavipes Isolated from Freshwater Environments"

_jof, 2025, doi:10.3390/jof11120857_

Round 1
Reviewer 1 Report (New Reviewer)
This manuscript describes the isolation, identification, and antimicrobial evaluation of secondary metabolites from the freshwater-derived fungus Trichoderma flavipes FBCC-F1632. The study is well-structured, following a logical progression from fungal isolation to compound characterization.
1.The introduction correctly states that freshwater fungi are understudied. However, the novelty of the study would be significantly strengthened by clarifying whether the isolated compounds (cordyol C, diorcinol, violaceol I/II, tryptophol) have been previously reported from other strains of Trichoderma flavipes or if this is the first report from this specific species. This distinction is crucial for defining the contribution.
2.The methodology for the compound separation and purification and structure analysis assays needs more detail. Please specify the standard guidelines followed the MIC , HPLC and NMR and describe the bacterial strains used (e.g., ATCC numbers). Furthermore, it is essential to include quality control measures, such as testing a known antibiotic (e.g., vancomycin) alongside the compounds to validate the assay results.
1. The abstract and results mention compound F1632-6 (violaceol II), but its specific MIC/MBC values against the two test bacteria are not provided. These data are critical and must be included for a complete comparative analysis. 2. The manuscript includes figures (e.g., chromatograms, inhibition zones), please ensure they are of high resolution and clearly labeled. Like in Figure 4,what do they mean for the circle piece of paper in each plate. and Figure 5. ,there are two vertical axes in Agraph, ,what do they mean? In B graph, what is the purple part ? the graph is not clear and what does the x-axis and y-axis represent?
Author Response
Please see the attachment.

Reviewer 2 Report (New Reviewer)
The manuscript (jof-3900281) reports the antimicrobial compounds from Trichoderma flavipes isolated from freshwater environments. Five known compounds, F1632-1 (cordyol C), F1632-2 (diorcinol), F1632-3 (vi- 21olaceol I), F1632-4 (tryptophol), and F1632-6 (violaceol II), were identified, these compounds exhibited moderate antibacterial activity , particularly against Staphylococcus aureus and Bacillus cereus. Overall, this article lacks novelty in terms of the microbial strains, compound structures, and antibacterial activity discussed, So I'm very sorry to think that the article is not suitable for publication in this magazine.
- P8 line 224, HMQC is not belong to Long-range correlations.
- Page 10, figure 6, you can just draw the key HMBC correlations that are crucial for determining the structures.
Author Response
Please see the attachment.

Reviewer 3 Report (New Reviewer)
This study investigates the antibacterial activities of metabolites produced by the fungus Trichoderma flavipes isolated from freshwater environments in Korea. Separation and purification of active fractions were performed, as well as structural analysis of their compounds using NMR techniques. This research fits into new trends in the development of antimicrobial compounds from natural sources, which is a consequence of the increasing resistance of bacteria to antibiotics.
Here are some suggestions for the manuscript:
- Lines 170-187: Have S. aureus and B. cereus strains been tested against a standard set of antibiotics, such as vancomycin, ciproflaxin, chloramphenicol, etc. Is S. aureus CCARM3089 a methicillin-resistant strain? What are the inhibition zone values ​​for the antibiotic standards used for comparisons?
- Lines 229-234: Would it be clearer if the names of the detected compounds were written in Figure 6b: Cordyol C, Diorcinol, Violaceol, etc.
- In the Discussion section: There is no data on how effective the isolated compounds are compared to potential antibiotics used against the strains studied.
Author Response
Please see the attachment.

Reviewer 4 Report (New Reviewer)
The authors presented the results of a promising study aimed at searching for and identifying new high-molecular organic compounds with antimicrobial and antifungal activity. Five metabolites of the fungus Trichoderma flavipes FBCC-F1632 were isolated and chemically characterized, revealing compounds with antibacterial, plant-derived antifungal, antimalarial, and even cytotoxic properties. The presented data represent the beginning of a promising study and offer hope that in the near future, the compounds described in this manuscript will be further characterized in more detail regarding their effects on higher organisms. Due to the growing antibiotic resistance of pathogens to existing antibiotics, the search for and discovery of new antibiotics with high activity against pathogens is of great importance for the development of modern pharmacology. The text is written in accessible language and its meaning is clear. The manuscript contains minor flaws, which I would recommend the authors strive to correct.
Methods:
Subsection 2.3. Lines 94-95: The authors noted that Staphylococcus aureus CCARM3089 and Bacillus cereus CCARM0120 strains were used to test the antibacterial activity of Trichoderma flavipes FBCC-F1632 fungal extract. It would be desirable to supplement this section with the main characteristics of these strains, indicating the reasons for their use, for example, their unusual resistance to antibiotics.
Results:
Lines 179-180, Figure 4 caption: The authors listed panel (A) but omitted section (B). Additionally, the word "and" was repeated. This should be corrected.
Lines 228-229, Figure 6: "Bold lines" mentioned in the figure caption are poorely differentiated and should be drawn more prominently.
Author Response
Please see the attachment.

This manuscript is a resubmission of an earlier submission. The following is a list of the peer review reports and author responses from that submission.
Round 1
Reviewer 1 Report
This paper focuses on the research of antimicrobial compounds from Trichoderma flavipes FBCC - F1632 isolated from freshwater environments. The research topic is of significant scientific importance and potential application value. It provides valuable information for the exploration of new antimicrobial agents. However, there are a few major queries I would like to discuss with the author.
Q1: L67, Why this site was chosen
Q2: L73, Why is it 15℃ and not 4℃
Q3: L76, How many strains were isolated and identified? Why was the strain FBCC-F1632 selected?
Q4: L95, Why were the two strains Fusarium solani KACC47794 and Phytophthora capsici 95KACC40470 selected
Q5: L133, Please improve the method of morphological identification
Q6: L155, Please supplement the data of fungal morphological identification, which is incomplete
Q7:L198, Why are Fusarium solani KACC47794 and Phytophthora capsici KACC40470 mentioned in the previous material method, while another three strains are used here?
Q8:L339,Please add the method of bacteriostasis test?
Q9:Why are fungi used in the first half of the test and bacteria used in the second?
Q10:According to the experimental results, please readjust the order of discussion parts and increase the mechanism discussion.
This paper focuses on the research of antimicrobial compounds from Trichoderma flavipes FBCC - F1632 isolated from freshwater environments. The research topic is of significant scientific importance and potential application value. It provides valuable information for the exploration of new antimicrobial agents. However, there are a few major queries I would like to discuss with the author.
Q1: L67, Why this site was chosen
Q2: L73, Why is it 15℃ and not 4℃
Q3: L76, How many strains were isolated and identified? Why was the strain FBCC-F1632 selected?
Q4: L95, Why were the two strains Fusarium solani KACC47794 and Phytophthora capsici 95KACC40470 selected
Q5: L133, Please improve the method of morphological identification
Q6: L155, Please supplement the data of fungal morphological identification, which is incomplete
Q7:L198, Why are Fusarium solani KACC47794 and Phytophthora capsici KACC40470 mentioned in the previous material method, while another three strains are used here?
Q8:L339,Please add the method of bacteriostasis test?
Q9:Why are fungi used in the first half of the test and bacteria used in the second?
Q10:According to the experimental results, please readjust the order of discussion parts and increase the mechanism discussion.